# Vertical Cylinder-to-Lamella Transition in Thin Block Copolymer Films Induced by In-Plane Electric Field

**DOI:** 10.3390/polym13223959

**Published:** 2021-11-16

**Authors:** Alexey S. Merekalov, Yaroslav I. Derikov, Vladimir V. Artemov, Alexander A. Ezhov, Yaroslav V. Kudryavtsev

**Affiliations:** 1Topchiev Institute of Petrochemical Synthesis, Russian Academy of Sciences, 119991 Moscow, Russia; alexis@ips.ac.ru (A.S.M.); derikoff@ya.ru (Y.I.D.); alexander-ezhov@yandex.ru (A.A.E.); 2Shubnikov Institute of Crystallography, Federal Scientific Research Centre “Crystallography and Photonics”, Russian Academy of Sciences, 119333 Moscow, Russia; artemov@crys.ras.ru; 3Faculty of Physics, Lomonosov Moscow State University, 119991 Moscow, Russia; 4Frumkin Institute of Physical Chemistry and Electrochemistry, Russian Academy of Sciences, 119071 Moscow, Russia

**Keywords:** block copolymers, microphase separation, electric field, solvent vapor, atomic force microscopy

## Abstract

Morphological transition between hexagonal and lamellar patterns in thin polystyrene–*block*–poly(4-vinyl pyridine) films simultaneously exposed to a strong in-plane electric field and saturated solvent vapor is studied with atomic force and scanning electron microscopy. In these conditions, standing cylinders made of 4-vinyl pyridine blocks arrange into threads up to tens of microns long along the field direction and then partially merge into standing lamellas. In the course of rearrangement, the copolymer remains strongly segregated, with the minor component domains keeping connectivity between the film surfaces. The ordering tendency becomes more pronounced if the cylinders are doped with Au nanorods, which can increase their dielectric permittivity. Non-selective chloroform vapor works particularly well, though it causes partial etching of the indium tin oxide cathode. On the contrary, 1,4-dioxane vapor selective to polystyrene matrix does not allow for any morphological changes.

## 1. Introduction

Owing to the immiscibility of chemically different monomer sequences, block copolymers can self-assemble into various nanostructures in melts and solutions. This tendency, which has been extensively investigated for the last 50 years [1,2], is now widely used in industry for the fabrication of semiconductor, photovoltaic, membrane, medical, and other functional nanodevices [3,4,5,6,7,8,9,10,11]. Application of patterned block copolymer thin films as nanoimprint lithography templates attracts most practical interest and importance [12]. Morphology of the microphase-separated state in a film is determined by the copolymer composition, monomer unit interactions, and surface effects from the substrate and atmosphere. Directed self-assembly of block copolymers is a comprehensive concept that encompasses a variety of techniques implemented to obtain large defect-free high aspect ratio patterns [13,14] in a technologically reasonable time [15]. The ways to effectively tailor an emerging nanodomain structure exploit the selectivity of copolymer blocks toward film surfaces (exposure to solvent vapors [16,17,18,19] and chemo- or grapho-epitaxy [20,21]), organic or inorganic dopants [22,23,24,25], and long-range shear, thermal, light, electric, or magnetic external fields [26,27,28,29,30,31,32].

The possibility of using a DC electric field for the orientation of block copolymer lamellae along the field lines was first demonstrated by Amundson et al. in the early 90s [33,34,35]. Since then, the electric field effects have been investigated rather intensively experimentally [36,37,38,39,40,41,42,43,44,45,46,47,48], by analytical/numerical theory [49,50,51,52,53,54,55,56,57,58,59,60], and via computer simulations [61,62,63]. It was demonstrated that the electric field has a strong impact on the conditions of order-disorder and order-order transitions in block copolymer melts, solutions, and thin films and can be considered as a precision tool to adjust the morphology of microphase-separated states on a nanometer scale. Its nanoimprint applications are however still limited, as strong fields that are needed to get rid of the structural imperfections can cause a dielectric breakdown of the polymeric material [64].

Electric switching between different orientations of highly ordered lamellar or cylindrical morphologies in a thin block copolymer film is also a problem of undoubted practical interest. Previous studies reviewed by Pester et al. [31] reveal the existence of the critical field needed to reorient microdomains and different kinetic pathways that depend on the degree of immiscibility of copolymer blocks. For example, if lying (horizontal) cylinders are expected to rotate by 90°, they first disintegrate into particles at weak segregation or form a disordered continuous structure at strong segregation [53]. In any case, there is no long-range order in the transient state.

One should also note that the most practically important microstructure, namely standing (vertical) cylinders which connect two surfaces of a thin polymer film, was not studied under in-plane electric fields. In the present paper we address this situation and demonstrate that an unusual realignment scenario preserving minor component channels between the film surfaces can be realized. To this end, we (i) use solvent vapor to maintain the conditions favorable for vertical cylinders and to enhance their mobility throughout the whole process of rearrangement and (ii) dope the copolymer with Au nanorods which are predominantly localized within the cylindrical domains to increase their dielectric permittivity. It should be mentioned that the combinations of electric field with other external factors such as solvent vapor [65,66], shear field [67,68], and graphoepitaxy patterning [69] were successfully used in the literature to promote orientation of the lamellar microstructure, and the addition of nanoparticles was shown to lower a threshold value of the electric field needed for the domain ordering [44]. Our study implements both approaches to reveal how the initially well-defined hexagonal structure of standing block copolymer cylinders changes toward standing lamellar structure when a thin polymer film is placed into an in-plane DC electric field.

Among various diblocks, copolymers of styrene and 2- or 4-vinyl pyridine are very popular because of their synthetic availability via controlled polymerization techniques [70,71] and rather high block incompatibility [72], which can be further increased by exploiting the chelating ability of pyridine ligands toward metal ions and nanoparticles [73]. As a result, high-resolution contrast between microphase-separated domains can be attained [74,75]. As the ordering in such copolymers is sensitive to electric fields [44,46,48], our choice of polystyrene–*block*–poly(4-vinylpyridine) copolymers for the present study becomes understandable. After preliminary screening of several solvents, such as toluene, chloroform, dimethylformamide, dioxane, tetrahydrofurane, and ethanol, for vapor annealing, we chose chloroform as a good solvent for both styrene and 4-vinyl pyridine blocks and 1,4-dioxane as a selective solvent providing highest contrast between the microdomains upon vapor annealing.

## 2. Materials and Methods

Tetrachloroauric(III) acid trihydrate (HAuCl_4_·3H_2_O, 99%) was supplied by Aurat (Russia). Polystyrene–*block*–poly(4-vinylpyridine) (PS–P4VP, *M_n_*(PS) = 330 kg mol^−1^, *T*_g_(PS) = 100 °C, *M_n_*(P4VP) = 125 kg mol^−1^, *T*_g_(P4VP) = 153 °C, *Ð* = 1.18) prepared by living anionic polymerization was purchased from Polymer Source (Dorval, Canada). All other reagents and solvents (99%) used for Au nanorod (AuNR) synthesis were obtained from Sigma-Aldrich and used as received. Glass substrates covered (using PET photomask) by custom patterned indium tin oxide (ITO) electrodes (sheet resistance 15 Ω/sq) with a rectangular gap of 100 μm wide were purchased from Xin Yan Technology Ltd. (Kowloon, Hong Kong).

Hydrophilic AuNRs stabilized with cetyltrimethylammonium bromide were synthesized according to the seeded growth method [76,77,78] with certain modifications [79]. All the glassware was cleaned by aqua regia and rinsed with distilled water prior to the experiments. To prepare PS–P4VP/AuNR composite, 6 mg of PS–P4VP were dissolved in THF/ethanol (0.6 mL/1.4 mL) mixture. Ethanol was added to THF to increase the solubility of P4VP domains. 1.07 mL of AuNR aqueous sol (0.6 mg Au) was centrifuged at a rate of 15,000 rpm for 8 min, colorless phase was discarded. The precipitant was redispersed in 140 μL of water and slowly added to the polymer solution under vigorous stirring. Red to violet cloudy solution then was ultrasonicated for 1 h, left for 72 h undisturbed, ultrasonicated again for 20 min and dried by Ar flow followed by a short vacuum drying. The dark red solid was dispersed in 0.9 mL of chloroform. The resulting clear deep red to violet liquid was ultrasonicated for 20 min and filtered through the 0.2 μm polytetrafluoroethylene (PTFE) filter. The composite obtained in this way contains 7 wt% of AuNRs as measured by UV-vis absorption spectroscopy using a USB2000 spectrometer (Ocean Optics, Dunedin, FL, USA) in chloroform.

For the film preparation 100 μL of 0.5% PS–P4VP/AuNR composite or pure PS–P4VP copolymer dispersion in chloroform was spin coated to the ITO-covered glass substrate at 3000 rpm for 30 s. The substrate was placed into a polypropylene box in a PTFE cell with a pair of electrical contacts attached to the ITO electrodes. A glass vial with excess of a solvent (1,4-dioxane or chloroform) was placed near the holder and the box was sealed to be saturated with vapors. 800 V DC electric field was applied to the substrate to provide the field strength of 8 kV cm^−1^, which was kept at a room temperature for 24 h. Parallel experiments were carried out without electric field.

Atomic force microscopy (AFM) measurements were made using a scanning probe microscope NTEGRA Prima (NT-MDT, Moscow, Russia) operated in a semi-contact mode. The peak-to-peak amplitude of the “free air” probe oscillations was from 20 to 25 nm. Silicon cantilevers PointProbe^®^ Plus AFM Tips PPP-NCH-20 (Nanosensors, Neuchatel, Germany) and NSG01 “Golden” series cantilevers for semi-contact mode (NT-MDT, Moscow, Russia) were used. Measurements carried out on the same samples using different cantilevers led to the same results. Image processing was performed using the Image Analysis software (NT-MDT, Moscow, Russia). Characteristic parameters of the observed morphologies are shown in in the Appendix A.

Scanning electron microscopy (SEM) and energy-dispersive X-ray spectroscopy (EDS) measurements were carried out using a JSM-7401F scanning electron microscope (JEOL, Tokyo, Japan) equipped with a cold cathode field emission gun and a JED-2300 Analysis Station (JEOL, Tokyo, Japan) for EDS equipment. Secondary electron images were acquired using the lower secondary electron detector at an accelerated voltage of 1 kV and working distance of 8–9 mm without conductive coating. Accelerated voltage of 10 kV and probe current of 2.5 nA was used for EDS measurements.

For SEM measurements, the copolymer films annealed in chloroform vapors under electric field were stained in iodine vapor. To this end, the PS–P4VP and PS–P4VP/AuNR films were placed into a small sealed box with 0.2–0.3 g of solid iodine and left for 3 h at 60 °C. An excess of iodine was removed from the film by vacuum drying for 24 h.

## 3. Results

### 3.1. Set-Up Design and Sample Preparation

In thin diblock copolymer films, the period of a microphase-separated microstructure is comparable in order of magnitude with the film thickness. As a result, restructuring caused by external fields takes place on the same scale along and across the film surface. This creates opportunities for a smooth continuous transition between two different morphologies.

A schematic of our set-up in shown in Figure 1a. It includes a glass substrate with ITO semi-circular electrodes deposited by standard photolithography and connected to a DC voltage source. The set-up is mounted into a PTFE holder (Figure 1b) and placed in an airtight polypropylene box, optionally with a small open container filled with a liquid solvent, which can evaporate within the box until saturation is reached.

The electrodes facing each other are separated by a rectangular gap with a width of 100 μm (Figure 1c). Prior to voltage application, an 1,4-dioxane or chloroform solution of PS–P4VP diblock copolymer optionally doped with Au nanorods is poured onto the substrate using a spin coater. The solvent quickly evaporates and in a few minutes all the substrate appears to be covered with a diblock copolymer film, as evidenced by AFM imaging (Figure 1d). If the copolymer contains Au nanorods, they prefer to be located within P4VP domains, as demonstrated by us previously [80,81].

AFM scanning of the substrate surface across the gap in the absence of polymer coating allows one to measure that the ITO electrodes are 85 nm thick (Figure 1e). After spin-coating, the substrate is fully covered with a copolymer film. Its mean thickness (40 nm) can be found via scanning across the scratch made with a copper knife (Figure 1f; more details can be found in the Appendix A). Scanning the native film surface between the electrodes demonstrates that upon the coating the step height at the electrode edge is kept near 85 nm, while it becomes less steep (cf. Figure 1g,e). Thus, the polymer film thickness over the electrodes and over the gap is nearly the same. Annealing the diblock copolymer in the chloroform vapor without applying an electric field results in the formation of a standing cylinder structure in the diblock copolymer film over both the electrode and glass substrate within the gap (Figure 1h). Cylinders are formed of P4VP blocks (125 kg mol^−1^) and the surrounding matrix of PS blocks (330 kg mol^−1^). All further experiments were performed starting from this microphase-separated state.

### 3.2. Annealing Experiments

Figure 2 presents AFM scans of the annealed films supplemented with their 2D Fourier transform images shown in the insets. One can see that apparent changes of the film surface morphology essentially depend on the conditions of annealing.

The highest contrast picture is observed in Figure 2a, when a pure PS–P4VP film is annealed in the presence of 1,4-dioxane vapor, which is selective to PS blocks. The vapor annealing experiment is terminated by opening the polypropylene box and exposing the film to the atmosphere. The swollen PS matrix shrinks in the vertical direction, while the non-swollen P4VP cylinders keep their shape and get some freedom to adjust their positions. As a result, the height difference which provides contrast to the topographical AFM image markedly increases. For the PS–P4VP/AuNR composite (Figure 2e), the contrast is even more pronounced since some of the P4VP cylinders contain large (29 nm × 9 nm) AuNRs. However, the same factor deteriorates the long-range hexagonal order of standing cylinders, which is clearly evidenced from the blurred circle at the 2D Fourier transform image (cf. insets in Figure 2a,e).

One can see less height difference between the cylinders and matrix when the film is annealed under chloroform vapor (Figure 2c). This solvent is only weakly selective relative to the copolymer blocks [82] so that swelling and subsequent drying affect the whole film. Presence of AuNRs somewhat enhances the contrast but completely destroys the hexagonal structure of the composite (Figure 2g).

Now consider the results of the solvent vapor annealing for 24 h in the presence of a 800 V in-plane electric DC field. In the case of dioxane, virtually nothing happens both with the pure diblock copolymer film (Figure 2b), and with the composite film (Figure 2f). Much more interesting results are obtained in the experiments with chloroform, where the copolymer (Figure 2d) and, especially, the PS–P4VP/AuNR composite films (Figure 2h) undergo drastic morphological changes. Dots depicting the ends of the P4VP standing cylinders line up and partially merge into parallel stripes so that the hexagonal order is replaced with the lamellar one.

The most-ordered film areas before and after application of the DC field for 24 h are shown in Figure 3.

The initial film (Figure 3a) demonstrates nearly perfect hexagonal structure with a period of 78 ± 14 nm, whereas the electric field application results in a mixed cylinder/lamella morphology strongly oriented along the field strength vector both in the pure diblock copolymer (Figure 3b) and composite (Figure 3c) films. In the latter case, the contrast between microphase-separated domains is higher, and the film surface is considerably rougher.

Coexistence of lamellas and standing cylinders in locally different proportions becomes evident in less ordered areas of the film as shown in Figure 4a,b. Although it is clear that lamellas appear via lining up cylinders and their further merging, up to 5 times longer experiments do not noticeably change the picture attained after 24 h of annealing (Figure 4c). 

Thin film ordering by the electric field on the tens of microns scale can be detected not only by AFM, but also by SEM (Figure 5). Image contrast is enhanced by exposing the sample to iodine vapor, which readily forms a complex with a pyridine ring in vinyl pyridine units [83] (Figure 5b,c). Au nanorods, which are also selective to P4VP blocks [80,81], are best visible with SEM on the standing cylinder structure (Figure 5d), which remains intact in the field-free areas of the film. It is seen that only few cylinders contain nanoparticles, however, merging cylinders to lamellas under the DC field should increase the fraction of P4VP domains with Au NRs. 

Extensive AFM scanning over the whole interelectrode space demonstrates that the cylinder-to-lamellar transformation proceeds to a highest extent in the central area of the gap (Figure 6b–d), while closer to the electrodes (Figure 6a,e) standing cylinders persist and even their lining up is not much pronounced. The continuous picture of morphological changes over 15 μm on approaching the anode is shown in Figure 6f.

The near-cathode area pointed to by the AFM tip in Figure 7a includes morphologically different areas visible in the overview topographic scan presented in Figure 7b. One can clearly see the electrode edge characterized by a rather high roughness amplitude of ~5 μm, which reflects the precision of electrode deposition provided by a standard poly(ethylene terephthalate) photomask. Whereas this roughness can lead to significant local inhomogeneity of the electric field, the edge pattern does not correlate with the striped structure formed in the interelectrode space on the right side of Figure 7b. Taking into account that the interelectrode distance of 100 μm is much larger that the electrode roughness, it is unlikely that the latter can be the reason for the apparent morphological changes in the copolymer film.

A typical strip of 2–4 μm wide is formed by dozens of lines (Figure 7c) with a period of 103 ± 13 nm, each line consisting of aligned and partially or fully merged P4VP domains (Figure 7h). Merging is much less pronounced between the stripes, where the initial cylindrical structure formed by the shorter copolymer blocks is still well discernible (Figure 7g). The average distance between neighboring stripes is ca. 7 μm. Almost no structural changes occur in the regions where the copolymer film is deposited over the cathode (Figure 7d) including its very edge (Figure 7e,f). At the same time, a number of particles up to 0.5 μm size are visible on the cathode and they also form a surface layer that follows the contour of its edge (Figure 7b). As seen from the optical images in Figure 7a, the near-anode region is darkly outlined. It corresponds to a ledge of ca. 1 μm height formed at the edge of this electrode, which makes exploration of the area with AFM hardly possible.

Thus, our experiments demonstrated that the annealing of a PS−P4VP diblock copolymer or a PS−P4VP/AuNR composite film under the in-plane DC electric field in the presence of chloroform vapor causes two phenomena: (i) reordering of the microphase-separated morphology in the polymer film between the electrodes, and (ii) mass transfer in the system, apparently of electrochemical origin. These effects will be discussed in the next section.

## 4. Discussion

### 4.1. Polymer Film Morphology

The possibility of using an electric field to control block copolymer morphology comes from the dielectric contrast between microphase-separated domains. In the strong segregation regime, which is realized in our case, the domains correspond to almost pure PS and P4VP components with the static dielectric permittivities ε_P4VP_ = 3.4 and ε_PS_ = 2.6 [84]. If 1,4-dioxane vapor (ε_d_ = 4.8) is present during the film annealing, it mainly penetrates PS domains thus effectively decreasing the dielectric contrast. This could be the reason why no domain reordering is detected in that case (Figure 2c,g). On the contrary, chloroform (ε_cl_ = 4.1) is a less selective solvent, which nearly equally interacts with PS and P4VP blocks thus preserving the difference between their permittivity values and increasing the mobility of polymer chains. As a result, the microphase-separated morphology can be effectively tailored with a strong electric field (Figure 2d,h).

Doping by Au nanoparticles was reported to markedly increase the low-frequency dielectric permittivity of poly(vinylidene fluoride) [85], vinylidene fluoride-trifluoroethylene-chlorofluoroethylene terpolymer [86], and SU-8 epoxy novolac resin [87], being attributed to the polarization of conductive particle interfaces within the dielectric polymer matrix, known as the Maxwell–Wagner–Sillars effect [88]. Since in our case the AuNRs are selective to the P4VP blocks [80,81], they increase the permittivity of the corresponding domains thus making the dielectric contrast with PS domains even higher. This can explain more pronounced morphological changes and growth of the surface roughness caused by the same electric field applied to the PS–P4VP/AuNR composite films (Figure 2h and Figure 3c) in comparison with the PS–P4VP pure copolymer films (Figure 2d and Figure 3b).

Negative electrostatic free energy always favors maximum effective capacitance, which corresponds to the situation when the domains are parallel to the electric field. Our observations demonstrate that this tendency is implemented by the formation of standing lamellas. Let us compare the effective dielectric permittivity ε characterizing cylindrical and lamellar morphologies of a diblock copolymer film using the effective medium approximation for anisotropic composites [89].

If inclusions with the (static) dielectric permittivity ε_1_ form fibers or layers within another material with the permittivity ε_2_, which are oriented along the DC electric field, then the effective permittivity reads ε_=_ = φε_1_ + (1 − φ)ε_2_, where φ is the volume fraction of the first component. This result, which describes the lamellar morphology in our case, can be also obtained via the simple capacitor analogy [90]. If, however, the inclusions in the form of cylindrical fibers are oriented perpendicular to the field, the effective permittivity has the form ε_⊥_= (1 + ∆εφ/(ε_2_ + ∆ε(1 − φ)/2))ε_2_, where ∆ε = ε_1_ − ε_2_ is the dielectric contrast between the inclusions and matrix. This formula corresponds to the standing cylinder morphology, where ε_1_ and ε_2_ describe the permittivity of P4VP and PS domains, respectively. The values of ε_1_ and ε_2_ can be effectively higher than ε_P4VP_ = 3.4 and ε_PS_ = 2.6 [84] due to the presence of Au nanoparticles and chloroform vapor, which we do not consider explicitly.

The complete transformation of the standing cylinder morphology into the lamellar one would lead to a relative change in the effective permittivity (ε_=_ − ε_⊥_)/ε_⊥_ = (∆ε/ε_2_)^2^φ(1 − φ)/(2 + (1 + φ)∆ε/ε_2_). Taking into account the block molar masses and densities for the PS–P4VP copolymer under study, one finds that the volume fraction of P4VP block equals φ = *V*_P4VP_/(*V*_P4VP_ + *V*_PS_) = 1/(1 + *V*_PS_/*V*_P4VP_) = 1/(1 + *M*_PS_*ρ*_P4VP_/(*M*_P4VP_*ρ*_PS_)) = 1/(1 + 330 × 1.15/(125 × 1.06)) ≈ 0.26. Therefore, for the pure diblock copolymer with ε_1_ = 3.4 and ε_2_ = 2.6, the relative permittivity gain due to the cylinder-to-lamellar transformation (ε_=_ − ε_⊥_)/ε_⊥_ ≈ 0.008, which is less than 1%. Chloroform is not very selective between PS and P4VP so that the copolymer swelling under its vapor should not considerably change the value of φ. Imagine that the minor phase permittivity ε_1_ increases up to the value of 4.1 characterizing pure chloroform, while ε_2_ remains equal to 2.6, which is typical of PS. Even in this unrealistic case the permittivity gain is no more than 2.3%. We know that the chloroform vapor treatment is enough to observe morphological transformations in the film, however, it is not surprising that the standing cylinders cannot be fully replaced with lamellas even at long annealing times. Adding 5% of Au nanoparticles increases the static polymer permittivity about 1.5 times [85,86,87]. If the value of ε_1_ in the PS–P4VP/AuNR composite attains 6.0, while ε_2_ persists at 2.6, then the above morphology transformation increases the electrostatic term in the free energy by 9% only.

Thus, the observed morphology rearrangement alters the electrostatic free energy quite insignificantly. It means that other contributions to the free energy of the system, which include entropic (conformational) and enthalpic (interfacial, due to the strong segregation of polymer blocks) terms also should not be markedly changed. A possible simple scenario of the domain relocation is depicted in Figure 8. At the first stage the hexagonally packed cylinders (Figure 8a) are oriented along the field (Figure 8b). Since the period of 78 ± 14 nm between the centers of the neighboring cylinders (Appendix A) is preserved, this is equivalent to the rotation of the whole sample by a certain angle. Then, the rows of cylinders are shifted apart from each other distorting the hexagonal structure (Figure 8c). In this state, the distance between neighboring rows is 101 ± 12 nm (Appendix A), while cylinders lined up within the same row are separated by a distance of 75 ± 14 nm (Appendix A). Finally, cylinders can merge to form a lamellar structure with a period of 103 ± 13 nm, which remains parallel to the electric field (Figure 8d). It is worth noting that the above changes are observed in both composite and pure diblock copolymer films.

The states depicted in Figure 8a,b are characterized by the same energy, therefore, the ordering of P4VP cylindrical domains along the electric field may be anticipated throughout the whole interelectrode gap. However, a transition to the state shown in Figure 8c and, especially, Figure 8d, is not thermodynamically beneficial, if a weak gain in the electrostatic energy is insufficient to overcome the repulsion between PS blocks that prevents merging of P4VP cylinders into lamellas. This speculation is in agreement with stable coexistence of the lined up cylinder and lamellar morphologies observed in our experiments. In this regard, it is appropriate to mention that mixed lamellar morphologies were theoretically predicted for the case when an out-of-plane DC field was applied to the surfaces of a compositionally symmetric diblock copolymer film [52] and experimentally detected for the above geometry with lamella-forming PS–P2VP films doped with PS-covered Au nanoparticles and heated above *T*_g_ of the both blocks [42].

### 4.2. Electrochemical Effects

Images of the near-cathode area in Figure 7 indicate the presence of solid particles after the DC electric field and chloroform vapor treatment of the copolymer or composite film. SEM/EDS analysis of these particles reveals (Appendix A) that they are indium chloride (InCl_3_) crystals. As indium is one of the elements constituting the electrode material (ITO), it is clear that the cathode in our set-up undergoes etching despite being covered with a polymer film. Chlorine atoms obviously come from chloroform vapor absorbed by the polymer film and then diffused toward the ITO substrate. However, a chemical interaction between CHCl_3_ and ITO is reported in the literature at the anode only, where chloroform can undergo oxidation into a CCl_3_ radical, which then binds to the ITO surface [91]. At the same time, ITO cathodes can be intensively etched by HCl either in liquid [92] or vapor [93] state. In aqueous media, the electrochemical routes of CHCl_3_ reduction are described [94,95], with Cl^–^ ions among the products. Without water, chloroform can transform to hydrochloric acid only via the interaction with atmospheric oxygen but, fortunately, this phosgene-producing route is very slow. Alternatively, CHCl_3_ can be electrochemically reduced up to methane via several stages of hydrodechlorination. The first stage of this reaction, which was demonstrated to proceed at metal cathodes of various nature [96], results in the elimination of chlorine atom, which in our case is able to interact with indium oxide component of the electrode material.

Therefore, it seems rather likely that, in the course of annealing, chloroform condensed from the vapor phase penetrates through the polymer film to the charged ITO electrode and reduces there, causing a partial transformation of indium oxide into indium chloride.

## 5. Conclusions and Outlook

In this study, we have investigated the possibility of changing the morphology of thin microphase-separated diblock copolymer films by simultaneous application of the in-plane DC electric field and solvent vapor at room temperature. We have found an interesting opportunity to rearrange standing cylinders of the minor component by lining them up in the field direction and to force their merging into standing lamellas. The resulting polymer film demonstrates mixed morphology, including the pristine hexagonal cylinder phase over the electrodes and coexistence of the standing lined-up cylinders with lamellas in the interelectrode space. The observed realignment effect is more pronounced, when the minor copolymer component is doped with Au nanoparticles. It opens new perspectives for the development of reconfigurable nanopatterns, in which the highly ordered copolymer state is preserved during morphological changes.

Simple analysis of the results has revealed that it would be more interesting to study the copolymer with a higher dielectric contrast between the domains, which could be enhanced by varying the chemical nature of the copolymer components and by using a solvent selective to the component with higher dielectric permittivity. The observed cathode etching demonstrates that the simultaneous use of a strong electric field and polar solvent can lead to unexpected side effects. An interesting extension of the present study could be related to the application of AC electric fields, which will lower the electric breakdown probability, eliminate effects related to ionic transport, and make it possible to adjust the dielectric contrast by changing the frequency of the applied field.

## Figures and Tables

**Figure 1 polymers-13-03959-f001:**
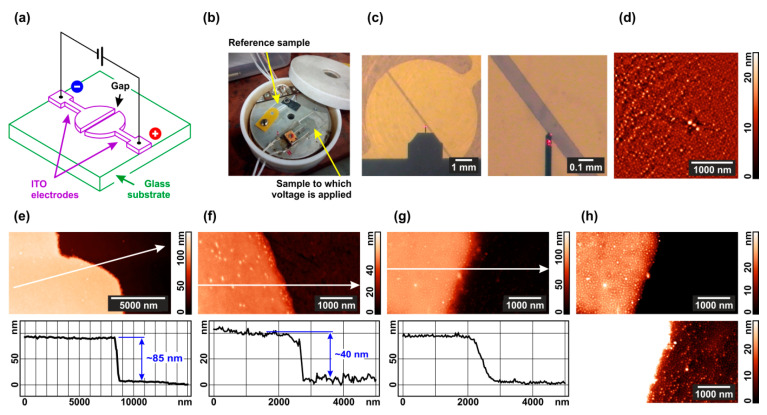
The experimental design and characterization of the samples: (**a**) Set-up schematic; (**b**) image of the cell for annealing under solvent vapor and electric field; (**c**) optical images of the ITO electrodes and the gap between them; (**d**) typical AFM image of the interelectrode space after spin-coating with a copolymer film; (**e**) AFM image and cross-section of the electrode edge before spin-coating; (**f**) AFM image and cross-section of a scratch made in the PS–P4VP film; (**g**) AFM image and cross-section of the electrode edge after spin-coating with a composite film; (**h**) typical topographic images of the composite film on ITO and glass after annealing in chloroform vapor without electric field. Height profiles in the Figure (**e**–**g**) are drawn along the arrows in the corresponding topographic images.

**Figure 2 polymers-13-03959-f002:**
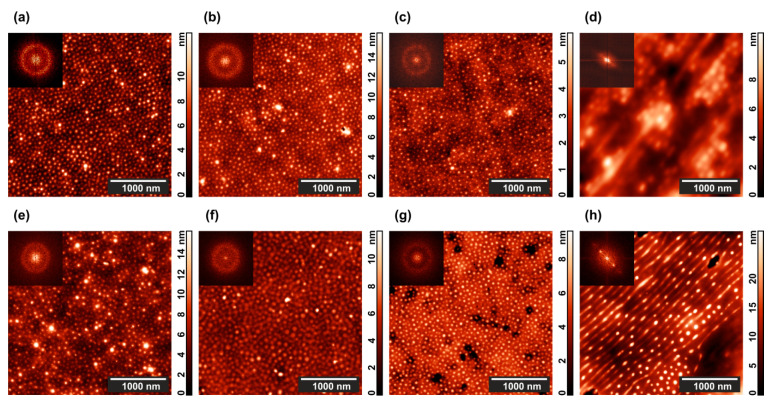
Typical AFM images of microphase-separated (top row) PS–P4VP copolymer and (bottom row) PS–P4VP/AuNR composite films after 24 h annealing in (**a**,**c**,**e**,**g**) 1,4-dioxane and (**b**,**f**,**d**,**h**) chloroform vapor (**a**,**b**,**e**,**f**) without and (**c**,**d**,**g**,**h**) with 800 V DC electric field. Two-dimensional (2D) Fourier transforms of the corresponding topographies are shown in the insets.

**Figure 3 polymers-13-03959-f003:**
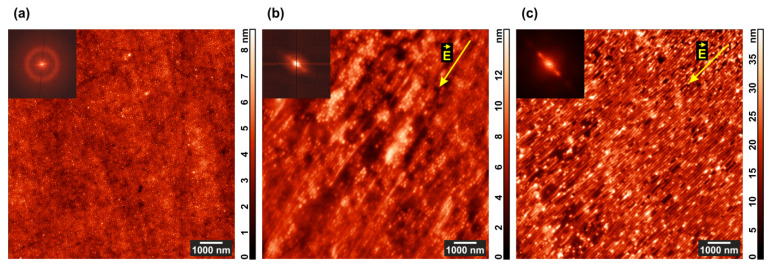
AFM images of the most-ordered areas in (**a**) PS–P4VP/AuNR film annealed under chloroform vapor; (**b**) PS–P4VP film annealed under chloroform vapor and DC field; (**c**) PS–P4VP/AuNR film annealed under chloroform vapor and DC field. 2D Fourier transforms of the corresponding topographies are shown in the insets.

**Figure 4 polymers-13-03959-f004:**
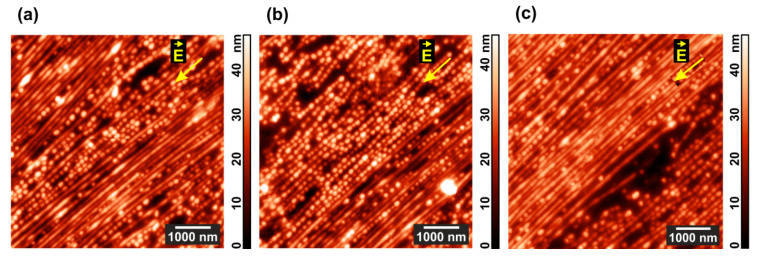
AFM images of the areas with a predominance of (**a**) horizontal and (**b**) vertical morphology in PS-P4VP/AuNR films annealed for 24 h under chloroform vapor and DC field. (**c**) The film surface after additional 96 h of the same treatment.

**Figure 5 polymers-13-03959-f005:**
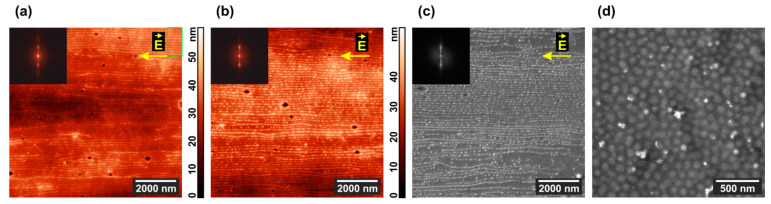
(**a**,**b**) AFM and (**c**,**d**) SEM images of the microphase-separated PS–P4VP/AuNR film annealed under chloroform vapor and DC field (**a**) before and (**b**–**d**) after its exposure to iodine vapor. The insets show fragments of Fourier transforms of the corresponding topographies.

**Figure 6 polymers-13-03959-f006:**
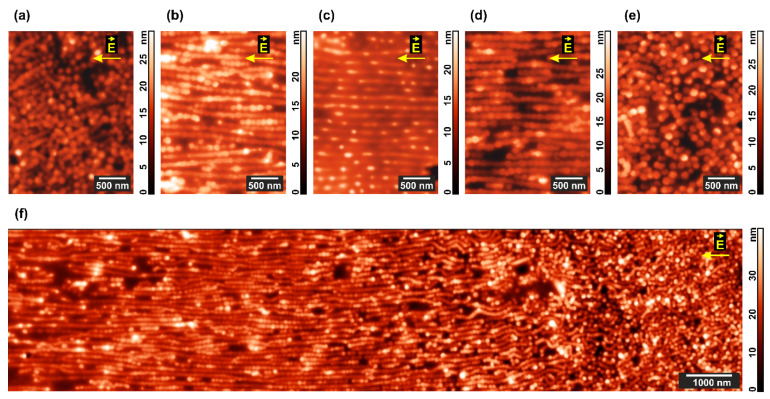
Domain structure of the microphase-separated PS–P4VP/AuNR film annealed under chloroform vapor and DC field. AFM images were taken from the (**a**–**e**) different internal areas on the way from cathode to anode and (**f**) large 15 μm area situated closer to the anode.

**Figure 7 polymers-13-03959-f007:**
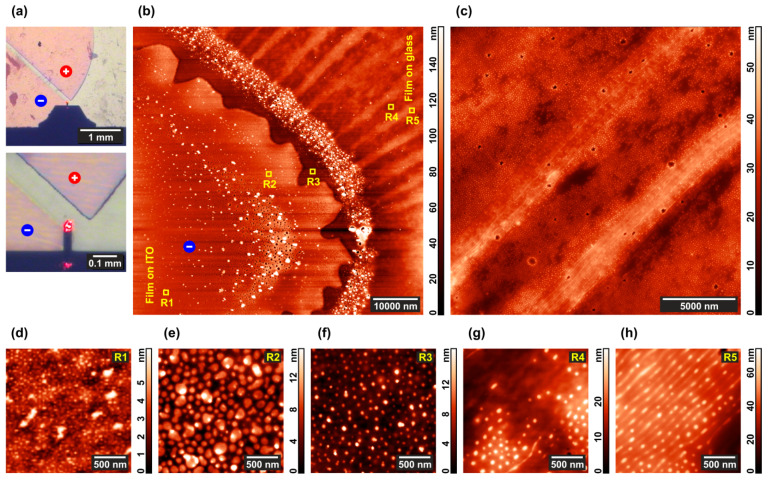
Details of the morphology in the microphase-separated PS–P4VP/AuNR film annealed under chloroform vapor and DC field. (**a**) Optical images of the electrodes; (**b**) AFM image of the near-cathode area of the film; (**c**) more detailed AFM image of the composite film between the electrodes; (**d**–**h**) AFM images of the typical domain structures in the R1-R5 regions marked in the image (**b**).

**Figure 8 polymers-13-03959-f008:**
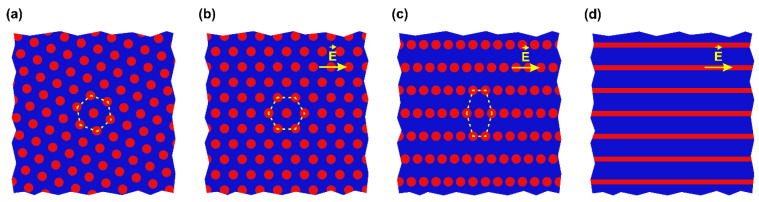
Schematic (top view) of the phase transformation in a strongly separated PS-P4VP film: (**a**) hexagonally packed standing cylinders; (**b**) field-oriented hexagonally packed cylinders; (**c**) lined up cylinders; (**d**) standing lamellas formed upon cylinder merging.

## Data Availability

The raw data presented in this study are available via the link https://drive.google.com/file/d/1rDVcSixEnvOhZmVr7Roxrp529i1r-Dga/view (accessed on 10 November 2021).

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
