# Peer review of "Vertical Cylinder-to-Lamella Transition in Thin Block Copolymer Films Induced by In-Plane Electric Field"

_polymers, 2021, doi:10.3390/polym13223959_

Round 1

Reviewer 1 Report

Dear Editor, in the present work vertical cylinder-to-lamella transition in thin block  copolymer films induced by in-plane electric field, have been prepared and studied.  It is an interesting work with new and advanced data and for this reason I propose to accept it for publication.

Some additional data could be added in Material section for the used copolymers.

It is not clear why this solvent was chosen.

Author Response

We are grateful to the referee for the interest in our paper and valuable comments.

Referee 1: Dear Editor, in the present work vertical cylinder-to-lamella transition in thin block copolymer films induced by in-plane electric field, have been prepared and studied.  It is an interesting work with new and advanced data and for this reason I propose to accept it for publication.

Some additional data could be added in the Materials and Methods section for the used copolymers.

Response: We purchased the PS-P4VP copolymer from a reliable manufacturer company Polymer Source (Canada) and used it as received. We have added the name of the synthetic method and values of Tg for both blocks to the Materials and Methods section (1st paragraph on page 2)

Referee 1: It is not clear why this solvent was chosen.

Response: In fact, we tried different solvents including their combinations and some of them led to interesting morphological transformations. However, to stay focused into the main effect, we discuss in the paper chloroform (non selective good solvent) and dioxane (selective solvent) only. We added an explanatory sentence in the very end of the Introduction section on page 2.

Reviewer 2 Report

The authors submitted the article entitled “Vertical Cylinder-to-Lamella Transition in Thin Block Copolymer Films Induced by In-Plane Electric Field”. I recommend that the paper could be accepted after minor revisions. My main comments and questions are as follows:

  1. In Fig 1, the authors should clearly describe how they prepared a sharp-edge coating films.
  2. In Fig 8 and relevant data, the authors should provide detailed statistic estimations of cylinder diameters.
  3. The authors should provide amounts/contents of AuNR on PS-P4VP.
  4. Although the manuscript provided numerous AFM data, the authors should further provide their application differences in different morphologies, such as electrochemical, sensing, or catalytic behavior…etc.
  5. Citations of several PVP related literatures are needed, such as Adv. Mater. 2015, 27, 4364; Polymer 2017, 121, 297; Macromolecules 2018, 51, 7491; Polymer 2021, 213, 123212; Nanomaterials 2021, 11, 2392.
  6. The authors should check the format of the references.
  7. English correction is recommended, such as missing subscripts in HAuCl4•3H2O, unifying h and hr…etc.

Author Response

We are grateful to the both referees for the interest in our paper and valuable comments.

Referee 2: The authors submitted the article entitled “Vertical Cylinder-to-Lamella Transition in Thin Block Copolymer Films Induced by In-Plane Electric Field”. I recommend that the paper could be accepted after minor revisions. My main comments and questions are as follows:

In Fig 1, the authors should clearly describe how they prepared a sharp-edge coating films.

Response: Sharp edges at the images come from ITO electrodes, which were deposited by standard photolithography using a photomask. When a polymer film is applied by spin-coating, the sharpness decreases (cf. Fig. 1e and 1g). Another source of the sharp edges is film scratching, which is intentionally performed to measure the film thickness if the film cannot be removed from the substrate, as in our case. To explain this to a reader, we added several words to the 3rd paragraph on page 4 and Figure S2 with some text in Supplementary Materials.

Referee 2: In Fig 8 and relevant data, the authors should provide detailed statistic estimations of cylinder diameters.

Response: Performing the statistical analysis of the morphological parameters is not easy in our case because the films are too thin to be removed from the substrate. However, we have analyzed several areas of the film and presented the results in Figure S1. The obtained estimates are mentioned in the main text as well, see pages 6, 8, and 10. Moreover, we modified the sketch in Figure 8 to make the scenario of the morphological evolution more clear.

Referee 2: The authors should provide amounts/contents of AuNR on PS-P4VP.

Response: The sentence “The composite obtained in this way contains 7 wt% of AuNRs as measured by UV-vis absorption spectroscopy in chloroform” was added to the end of the 1st paragraph on page 3.

Referee 2: Although the manuscript provided numerous AFM data, the authors should further provide their application differences in different morphologies, such as electrochemical, sensing, or catalytic behavior…etc.

Response: At the moment, the possible application areas of the observed effect are not fully clear. However, we believe that it could be used to manufacture reconfigurable nanopatterns. The corresponding sentence was added to the end of the 1st paragraph of the Conclusions and Outlook section on page 11.

Referee 2: Citations of several PVP related literatures are needed, such as Adv. Mater. 2015, 27, 4364; Polymer 2017, 121, 297; Macromolecules 2018, 51, 7491; Polymer 2021, 213, 123212; Nanomaterials 2021, 11, 2392.

Response: We introduced a new paragraph in the end of Introduction on page 2 explaining our choice of the copolymer and solvents for the present study. Six new refs [70]-[75] including four ones proposed by the reviewer were added to this paragraph. The last paper in Nanomaterials does not report anything about poly(vinyl pyridine).

Referee 2: The authors should check the format of the references.

Response: Done

Referee 2: English correction is recommended, such as missing subscripts in HAuCl4•3H2O, unifying h and hr…etc.

Response: Done